# The Effects of Verbal Encouragement during a Soccer Dribbling Circuit on Physical and Psychophysiological Responses: An Exploratory Study in a Physical Education Setting

**DOI:** 10.3390/children9060907

**Published:** 2022-06-17

**Authors:** Bilel Aydi, Okba Selmi, Mohamed A. Souissi, Hajer Sahli, Ghazi Rekik, Zachary J. Crowley-McHattan, Jeffrey Cayaban Pagaduan, Antonella Muscella, Makram Zghibi, Yung-Sheng Chen

**Affiliations:** 1Research Unit, Sportive Performance and Physical Rehabilitation, High Institute of Sports and Physical Education of Kef, University of Jendouba, EI Kef 7100, Tunisia; aydibilel18@yahoo.fr (B.A.); sahlihajer2005@yahoo.fr (H.S.); makwiss@yahoo.fr (M.Z.); 2Physical Activity, Sport and Health, Research Unit (UR18JS01), National Observatory of Sport, Tunis 1003, Tunisia; gaddoursouissi@yahoo.com; 3High Institute of Education and Continuing Training of Tunis, Virtual University of Tunisia, Montplaisir 1073, Tunisia; 4Research Laboratory: Education, Motricity, Sport and Health (LR19JS01), High Institute of Sport and Physical Education, Sfax University, Sfax 3000, Tunisia; ghazi.rek@gmail.com; 5Tanyu Research Laboratory, Taipei 112, Taiwan; 6Discipline of Sport and Exercise Science, Faculty of Health, Southern Cross University, Lismore 2480, Australia; zac.crowley@scu.edu.au; 7Faculty of Physical Culture, Palacký University, 77900 Olomouc, Czech Republic; jcpagaduan@gmail.com; 8Department of Biological and Environmental Science and Technologies (DiSTeBA), University of Salento, 73100 Lecce, Italy; antonella.muscella@unisalento.it; 9Department of Exercise and Health Sciences, University of Taipei, Taipei 111, Taiwan; 10Exercise and Health Promotion Association, New Taipei City 241, Taiwan

**Keywords:** verbal encouragement, physical education, Hoff circuit, enjoyment, mood, performance

## Abstract

Verbal encouragement (VE) can be used by physical education (PE) practitioners for boosting motivation during exercise engagement. The purpose of this study was to investigate the effects of VE on psychophysiological aspects and physical performance in a PE context. Twenty secondary school male students (age: 17.68 ± 0.51 yrs; height: 175.7 ± 6.2 cm; body mass: 67.3 ± 5.1 kg, %fat: 11.9 ± 3.1%; PE experience: 10.9 ± 1.0 yrs) completed, in a randomized order, two test sessions that comprised a soccer dribbling circuit exercise (the Hoff circuit) either with VE (CVE) or without VE (CNVE), with one-week apart between the tests. Heart rate (HR) responses were recorded throughout the circuit exercise sessions. Additionally, the profile of mood-state (POMS) was assessed pre and post the circuit exercises. Furthermore, rating of perceived exertion (RPE), traveled distance, and physical activity enjoyment (PACES) were assessed after the testing sessions. Furthermore, the CVE trial resulted in higher covered distance, %HRmax, RPE, PACES score, (Cohen’s coefficient d = 1.08, d = 1.86, d = 1.37, respectively; all, *p* < 0.01). The CNVE trial also showed lower vigor and higher total mood disturbance (TMD) (d = 0.67, d = 0.87, respectively, *p* < 0.05) and was associated with higher tension and fatigue, compared to the CVE trial (d = 0.77, d = 1.23, respectively, *p* < 0.01). The findings suggest that PE teachers may use verbal cues during soccer dribbling circuits for improving physical and psychophysiological responses within secondary school students.

## 1. Introduction

When designing teaching situations in team ball sports such as soccer, most physical education (PE) teachers rely heavily on diverse dribbling circuits [1]. Due to the integrated work (training with ball), these training circuits solicit various physical and cognitive performances in practitioners (students/players) [2,3]. Previous works in the literature have reported that soccer-specific training circuits are a commonly used training method that can simultaneously improve the physical, physiological, and technical aspects of players [4,5,6]. The set-up of training circuits depends on several factors that determine its success, namely distance, duration, the space provided, and the encouragement of the PE teacher [4]. 

Continuous stimuli provided by PE teachers is crucial to improve student’s physical and cognitive performances as well as their motivational beliefs [7]. It is believed that verbal encouragement (VE) from the practitioners can enhance intrinsic motivation [7,8,9,10,11], which clearly affects physical commitment and positive interaction with physical exertion and desire to do exercises [7,12]. This, in turn, leads to better technical, physical, and emotional responses [8,13,14].

Several studies about integrated training have reported the importance of the encouragement provided by the physical trainers on training intensity, expressed as perceived exertion and physiological responses [15]. In fact, the motivation resulting from integrated training sessions may be related to good mood state, positive physical pleasure, and vigorous exercise intensity [7]. For example, Selmi et al. [12] reported that encouragement cues result in optimal motivation and physical enjoyment, subsequently leading to enhancement of psychophysiological responses in participants during integrated exercises. Additionally, Selmi et al. [16] suggested that encouragement cues from practitioners ensures mood balance during integrated training in soccer players.

During PE sessions, VE provided by a course teacher is recognized as a form of external motivation that advances physical engagement [7]. However, limited information addresses soccer exercise training regarding the importance of VE on participant’s performance. To our knowledge, no study has tackled the effects of VE on psycho-physiological and emotional responses by utilizing a soccer specific intervention during technical circuits in school-based PE sessions. Given the positive benefits of encouragement on athletes’ motivation, as well as the potential influences of exercise intensity and positive affective responses, research to fill this gap in the literature is warranted. 

Therefore, the purpose of this study was to examine the impact of the PE teacher’s VE on physical performance and students’ perception of exertion, physical enjoyment, and mood while completing a training circuit with a ball (Hoff circuit). Using soccer-specific training drills in physical education sessions is important and beneficial for students as the use of these types of drills is educationally more effective and motivating than conventional drills. This specific intervention provides physical and psychological benefits due to positive feelings and high exercise intensity [17].

## 2. Materials and Methods

### 2.1. Participants

The study was approved by the research ethics committee of the High Institute of Sports and Physical Education of Kef, Tunisia (approval no. 011/2020). The experiments were conducted in accordance with the Declaration of Helsinki. Twenty male students enrolled in one secondary school in Tunisia were involved in this study (age: 17.68 ± 0.51 years; PE experience: 10.9 ± 1. years; height: 175.7 ± 6.2 cm; body mass: 67.3 ± 5.1 kg, %fat: 11.9 ± 3.1%). The participants were from the same study class and practiced two physical education sessions per week (Tuesday and Friday). All participants had no reported injuries or illnesses before or during the study. A researcher of our study group informed all risks and benefits associated with participation to participants, and their parents voluntarily agreed to participate in the research and gave written informed written consent after a detailed explanation about the objectives, procedures, and risks involved in the study. 

A priori power analysis was used to estimate the sample size (G*Power Version 3.1.9.4., Düsseldorf, Germany), based on the t test family (Means: difference between two dependent means). The analysis output showed that a sample size of 19 subjects would be sufficient to identify significant differences (effect size = 0.887, power (1 – β) = 0.95 with an actual power of 95.46 in this study.

### 2.2. Data Collection and Analysis

#### 2.2.1. Anthropometric Measurements

The participants’ height and body mass were measured using a standard protocol that the variability of measurement was within 0.2 kg and 5 mm. Height and body mass were measured to the nearest 0.1 cm and 0.1 kg with a digital scale (OHAUS, Florhman Park, NJ, USA), respectively. Four-sides skinfold thickness was used to determine the percentage of body fat (biceps, triceps, subscapular and suprascapular), using a calibrated Harpenden skinfold caliper (Holtain Instruments, Crosswell, Pembrokeshire, UK). A well-trained sports scientist conducted the anthropometric measurements in this study. The percentage of body fat (%body fat) was calculated via a validated method: %body fat = (495/body density) − 450; D = 1.1533 − (0.0643 × L) with D = body density (g/mL), and L = log of the total of the 4 skinfolds (mm) [18,19].

#### 2.2.2. Vameval Test

The Vameval test was conducted on a 200 m running track. The testing field was set with ten cones placed every 20 m at specific sites on the pitch following a preprogrammed auditory signal (i.e., a beep). The initial running speed was determined at 8 km.h^−^^1^ and subsequently increased by 0.5 km.h^−^^1^ every minute until exhaustion. The students controlled their running pattern between the cones. The test was terminated when a participant failed to maintain the running speed guided by the beep consecutive shuttles or felt they could not continue the run [12]. The herat rate (HR) was recorded via a Polar Team Sport System (Polar-Electro OY, Kempele, Finland). The highest HR average value over 5 s during the Vameval test was recorded as Vameval-HRmax. The reliability of the Vameval protocol to detect the maximal HR has been shown previously (a Cronbach’s α value of 0.83) [20].

#### 2.2.3. Heart Rate

During the Hoff circuit exercises, the HR was recorded via portable HR sensors (Polar Team Sport System, Polar-Electro OY, Kempele, Finland). The HR detection was recorded every 5 s. The participants frequently checked the position of the HR sensor throughout the exercise. Subsequently, the HR data were expressed as a percentage of Vameval-HRmax (%HRmax) and mean HR (HRmean). The %HRmax was calculated by the formula: %HRmax = (HRmean/Vameval-HRmax) × 100 [21].

#### 2.2.4. Rating of Perceived Exertion (RPE)

The internal load was measured directly after the Hoff circuit exercises. The Borg CR-10 RPE was used to assess the level perceived effort during the Hoff circuit exercise in this study [22]. The validity and reliability of this method were reported ed in previous studies [23]. The participants were familiarized with the RPE-scale in the first week of the study to maximize the accuracy of the answers.

#### 2.2.5. Profile of Mood States (POMS)

The Profile of Mood State (POMS) was used to evaluate mood state before and after the Hoff circuit exercises [24]. After a period of familiarization, the survey was administered to all participants 15 min before and 5 min after each training circuit to assess mood state [16]. The questionnaire consists of 37 items, rated on a 5-point Likert scale ranging (from 0, not at all to 4, extremely). The POMS assess six states: tension-anxiety, depression-dejection, anger-hostility, vigor-activity, fatigue-inertia, and confusion-bewilderment. The POMS subscales can be combined into a total mood score (TMD) score by adding the scores for the five negative states and subtracting the positive mood (vigor) score. Adding a constant of 100 to avoid negative numbers, [TMD = (Anger + Confusion + Depression + Fatigue + Tension) – Vigor + 100]. The Cronbach’s α of the POMS test ranged from 0.84 to 0.92 in the present study. The participants completed the POMS on papers outside where they completed the Hoff circuit exercise.

#### 2.2.6. Physical Enjoyment

The 8-item Physical Activity Enjoyment Scale (PACES) was completed for the assessment positive feelings associated with exercise training in this study [25]. The participants were asked to rate “how you feel at the moment regarding the exercise training you experienced” via a 7-points scale (ranging from 1, it’s very enjoyable to 7, it’s not fun at all). The final score was calculated by a sum of the 8 items scores. The score ranges from 8 to 56 points. The large number of PACES scores indicates high level of physical enjoyment [25]. The Cronbach’s α value of the PACES test was 0.89 in the present study. 

### 2.3. Procedure

Within the first week, the anthropometric characteristics were measured, and all participants performed the Vmaeval test [12] to obtain individual Hrmax (Figure 1). In the experimental days, two Hoff circuit sessions were performed separated by a one-week interval during regular PE sessions. The order of sessions was randomized and counterbalanced such that half of the participants completed the Hoff circuit exercise with VE (CVE) and the other half completed the Hoff circuit exercise without VE (CNVE) first and vise versa. Each subject performed one CVE trial and one CNVE trial.

The experimental sessions (Figure 2) were carried out on an artificial grass pitch in the morning physical education sessions (between 9:00 and 10:30 a.m.). The duration of each circuit exercise was 10 min. The HR was continuously monitored during each trial (CVE and CNVE). The distance traveled of the Hoff circuit exercise, RPE, and PACES were recorded immediately after the tests. Furthermore, the POMS was measured before and after each session. The participants reported the scales independently to avoid hearing the responses of their colleagues.

A standardized 12-min warm-up exercises was given to all participants (consisting of jogging, muscle coordination exercises, dynamic stretching, passing drills with ball, and ended with three 10 m sprints). No static passive stretching exercise was performed during the warm-up activities [26]. The participants only performed dynamic stretches before the experimental sessions [16]. Three minutes passive recovery was given after prior to the Hoff circuit exercise. All participants refrained from strenuous physical effort at least 48 h prior to experiments. All participants familiarized with the RPE, the PACES, the POMS, and the Hoff circuit during the familiarization week. A group of researchers and a PE teacher who collected all data.

### 2.4. Hoff Circuit Exercise

The Hoff circuit exercise was performed outdoor. The size of the pitch and the duration of the circuit training have been strictly implemented in other studies [19]. The track distance is 290 m. Participants move a soccer ball across the circuit by dribbling. The goal of the exercise was to do the maximum distance possible for 10 min. Each participant was informed the spent time at 5 min and at 9 min [19]. Five participants were grouped for the tests at the same time. Every minute, the testing signal is given to one participant. Thus, the researchers accounting the test had 4 min for starting the five participants and then switched to the halfway test signal that occurred in the successive minute for the first running participant. When the researchers indicated the halfway test signal for the fifth running participant (minute 9), at the same time began the last minute signal for the first running participant. The participants wore colored bibs that were always assigned in the same order to the running participants numbered 1 to 5 [27].

The participants were received VE from the PE teacher in the CVE trial, while they performed the test without VE in the CNVE trial. The PE teacher moved around the testing field while encouraging the participants (instruction such as Go Go Go, Again Again, Move, Again, Faster, More active, A bit of willpower, More effort, Courage and so on) [7,15]. The encouragement was spontaneous based on the behavior of the participants, according to his effort and movements. During the CNVE trial, the PE teacher stood next to the circuit and controlled the participants but did not provide VE cues.

#### Statistical Analyses

All data are presented as mean ± standard deviation (SD). The normality of the data was confirmed using the Kolmogorov-Smirnov test. Paired *T*-test was used to compare the Hrmax, Hrmean, physical variables (distance traveled), physical enjoyment (PACES score), RPE, and Hrmax. The Cohen’s coefficient (*d*) was used to determine the magnitude of differences between CVE and CNVE trials [28]. The sales of magnitude were considered trivial: *d* ≤ 0.20; small: 0.20 < *d* ≤ 0.50; medium: 0.50 < *d* ≤ 0.80; and large: *d* > 0.80 [29].

Regarding the POMS scores, a two-way Analysis of Variance (ANOVA) was used to study the effect of the “exercise method” (CVE and CNVE), “effort” (pre- and post- exercise) and their interaction (exercise method × effort) on the scores of the six subscales (depression, anger, confusion, fatigue, anxiety, and vigor) and the TMD. Partial Eta-Squared (η^2^) was used from two-way ANOVA outputs. When a significant interaction effect was found, the analysis was completed with a post-hoc Bonferroni test. The level of statistical significance was set at *p* < 0.05.

## 3. Results

### 3.1. Physical and Physiological Performance

Results presented in Table 1 indicate significant changes in the distance traveled (D), Hrmax, and RPE variables between the trials (all, *p* < 0.001). 

### 3.2. Physical Enjoyment

The PACES score is significantly higher (*p* < 0.001, *d* = 1.36, Large) in the CVE trial (39.95 ± 3.17), compared to that of CNVE trial (46.65 ± 6.41). 

### 3.3. Mood State

No significance of main effect (condition and time) and interaction on depression, anger and confusion scores were observed. These mood parameters were not significantly affected by either “effort” or “training method”. However, there was a significant main effect of training method on TMD, fatigue, and vigor and a significant main effect of effort on fatigue (Table 2). Additionally, a significant interaction effect was found for tension, fatigue and TMD (Table 1). In Figure 3, Bonferroni’s post-hoc comparisons shows that only CVE trial the TMD scores decreased significantly (from 102.65 ± 12.43 to 97.45 ± 11.16) and those for vigor increased significantly (from 16.95 ± 4.03 to 13.45 ± 5.92) while anxiety (from 1.65 ± 2.47 to 1.45 ± 2.33) and fatigue scores increased significantly for the CNVE trial (from 4.9 ± 2.51 to 8.6 ± 4.07).

## 4. Discussion

This study investigated the effect of the PE teacher’s VE on the HR, RPE, physical performance, physical enjoyment, and mood state of school male students during a soccer dribbling circuit exercise. The results of the present study were as follows: (1) the CVE condition increased distance traveled, %Hrmax and internal intensity to a greater extent than that of CNVE condition; (2) the physical enjoyment was greater after the CVE; and (3) The CVE condition resulted in positive mood state, compared to that of CNVE condition.

In term of internal factors, the result of this study indicated that the RPE was significantly greater during the CVE. This indicates that they performed the Hoff exercise with high intensity which results in a high solicitation of the cardiorespiratory demands [5,23]. During the CVE trial, the RPE is very hard (>7), compared to that of the CNVE trial. Therefore, VE from the PE teacher motivates students to exert great effort during the Hoff circuit exercise. This result suggests that the high level of perceived effort is associated with the effort produced by students during physical activity [15,16].

Our results have been confirmed by Rampinini et al. [15] showed that VE caused higher values in terms of RPE compared to the lack of encouragement during specific soccer training (4 vs. 4) by modifying the dimensions of the field (small, medium, and large). The results gave the following RPE values: small filed: 7.6 ± 0.5, 6.3 ± 0.5; middle filed: 7.9 ± 0.5, 6.6 ± 0.6; large field: 8.1 ± 0.5, 6.8 ± 0.5, respectively. For example, Brandes and Elvers [30] reported the effectiveness of VE in promoting physical effort, training exercise adherence and internal intensity. This result is in agreement with Sahli et al. [7], studied the effects of VE on psychophysiological aspects during small-sided games (SSG). They reported that HR and RPE values were significantly higher in SSG with VE. Additionally, Selmi et al., [12] compared the effects of VE during specific training soccer on physiological variables and RPE in youth soccer players. The authors reported that RPE score and %Hrmax were higher in exercise training with VE than in training exercise without VE.

Regarding physical performance, this study indicated that VE from the PE teacher has a positive impact on the distance traveled on the Hoff circuit exercise. The incentive variable leads to an improvement in running distance. These findings suggest that the PE teacher’s VE can motivate the male school students to put in a great level of commitment, physical engagement, and run a further distance. Overall, the results of the present study suggested that the CVE induced higher physical contributions to energy demands and exercise intensity, thus confirming the significance of encouragement in improving the physical condition of participants.

Affective responses to training activity are an essential characteristic of performance [5,31,32]. In addition to focus and engagement, mood and enjoyment can predict training adherence [7]. In the present study, we observed that the physical enjoyment score (PACES) measuring the CVE trial from the teacher was significantly higher than that measured after the CNVE trial, confirming with a recent study of Selmi et al. [16] who reported that players synchronized with VE had higher PACES scores during the SSG than that of the high intensity interval training (HIIT). Selmi et al. [12] claimed that reduced game exercises with VE led to greater physical enjoyment than games without verbal encouragement. Additionally, Sahli et al. [7] reported that PACES scores measured after PE sessions with the teacher’s encouragement were higher than that without encouragement. This result may explain that motivating factors may clarify a high level of PACES [7,33,34]. In this context, other studies have indicated that the students who are most motivated in PE sessions are those who have a higher level of enjoyment [7,12,35]. These findings also suggest that the physical enjoyment induced by the training methods may vary depending on the modality of exercise, the behavior of the PE teacher, the results, and the desire of the students [36]. Interestingly, our result was relatively supported by previous studies [5,12,15,34], indicating that the physical activity sessions with VE is also more enjoyable for students. Furthermore, these indications suggested that the encouragement factor was easily motivating and enjoyable for the students and could therefore be a more effective method for improving the psychological responses during exercise engagement.

Regarding the comparison of the mood’s response (POMS) between CVE and CNVE, the present study recorded that VE resulted in an improvement in the mood state. Our hypothesis was supported by the result showing VE from the PE teacher contributed to positive improvement in the mood state of the students.

The POMS is a common psychological measure for athletes during athletic training [7,12,18,32,37]. During motivational training exercises, participants generally reported positive changes in the mood state [7,16]. As shown in our study, the score of the POMS variables was positively altered with the presence of VE from the PE teacher. The anxiety score (11.36%) and fatigue (6.52%) were significantly reduced, while vigor was increased (26.02%). The significant increase in vigor and decrease in negative variables (anxiety and fatigue) of POMS in students leaded to a reduction in the TMD score. These findings are consistent with several studies that have investigated the effect of teacher/coach’s VE on psychological aspects during integrated training. Indeed, we have shown that VE ensures the stability of the mood state during intense exercises and reduced games (3 vs. 3) [16]. Thus, Sahli et al. [7] reported that training of SSG (4 vs. 4) with VE from the PE teacher resulted in a significant improvement in vigor, and a significant decrease in anxiety and TMD scores. The authors indicated that VE is a key determinant of positive mood in students. Along the same lines, Selmi et al. [16] claimed that lack of motivation in physical exercise leads to mood disturbance with higher fatigue, tension, and decreased energy. Additionally, Andradeet al. [38] stated that mood disorders decrease with the presence of motivation of the PE teacher.

Based on our findings, the absence of VE during a PE session causes an inability to concentrate and a lack of physical engagement which results in changes in the mood, the behavior, and the anxiety [7]. Lane and Terry [39] showed that vigor played a crucial role in positive improvement of the mood, while in our study we recorded a more remarkable increase in this condition. This outcome suggests that an increase in the vigor subscale score is likely to lead to a positive feeling during PE sessions. Other research has shown that the vigor that follows motivational training exercises causes improvements in mood, increased energy, and physical engagement [40,41]. It has been suggested to explain this that reduced vigor and increased TMD during physical activity sessions are associated with lack of motivation and physical pleasure which are associated with unpleasant psychological responses [7,16,31]. These results suggest that lack of motivation during physical activity causes negative feelings in students unlike motivational exercises [42]. They could be related to recent studies regarding the affirmation of the central role of VE in the intrinsic motivation of students [8,38,43].

During vigorous exercise such as the Hoff circuit exercise, motivational factors may explain the positive change in some of the different POMS scores. However, studies have reported that engaging in motivational sport training improves participants’ mood [43,44,45]. The present study indicated that the behavior of a PE teacher can influence student behavior, intensity of exercise, affecting psychophysiological and emotional responses, indicating that the performance is associated with a positive teaching style.

Several limitations should be considered when interpreting the current results. First, the study sample was small due to the difficulty of recruiting large numbers of homogeneous participants. Furthermore, the tests used only one circuit format and used only one average age of students. Finally, it would be interesting to relate these aspects with technical aspects such as time motion variables (zones of running intensity, etc.), since these aspects are important variables of performance.

This investigation has practical implications. To our knowledge, this investigation is the first to examine physical performance, perceived exertion, physical enjoyment, and mood state during a soccer-specific training circuit among students. The VE from a PE teacher can be considered as a necessary variable during PE sessions since it induces important physical aspects, and positive sensations. This is the reason why PE teachers should verbally encourage their students during specific physical activity sessions to increase the intensity of play, create positive emotional states, and improve the student’s performance.

## 5. Conclusions

The present study indicates that the RPE, the physical enjoyment, the distance traveled, Hrmax and the positive mood are higher during the Hoff circuit with the PE teacher’s VE in comparison with Hoff’s circuit without VE. The VE is suggested as an effective intervention to improve exercise intensity, physical performance, mood state, and physical enjoyment when a soccer-specific exercise is used during PE sessions. The findings demonstrated that PE teachers should utilize VE to improve the motivation, physical engagement, and commitment in school students. Future investigations examining VE should be conducted during other exercise training or sports game (individual or collective) of the PE session. Other aspects (i.e., physical, physiological, technical, and emotional aspects) and students with different fitness levels, gender, and age should be further explored to extend the applicability of the results.

## Figures and Tables

**Figure 1 children-09-00907-f001:**
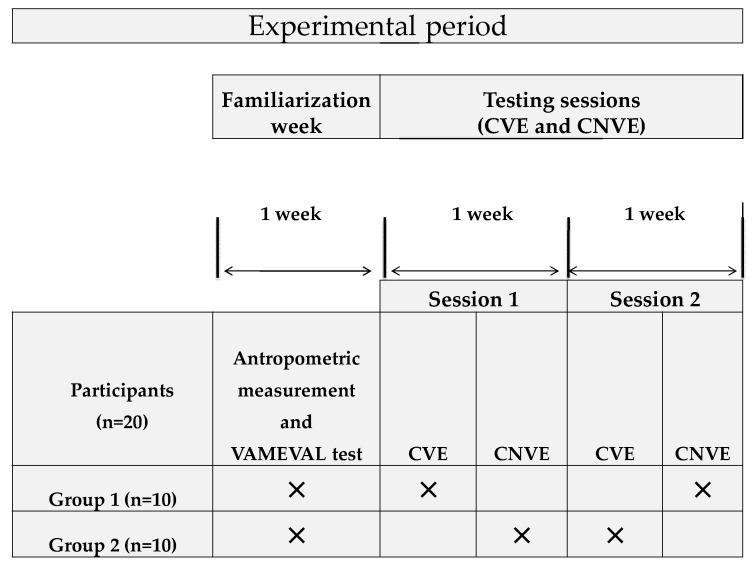
Representative diagram of the experimental protocol. n: number of participants; CVE: circuit exercise with verbal encouragement; CNVE: circuit exercise without verbal encouragement.

**Figure 2 children-09-00907-f002:**
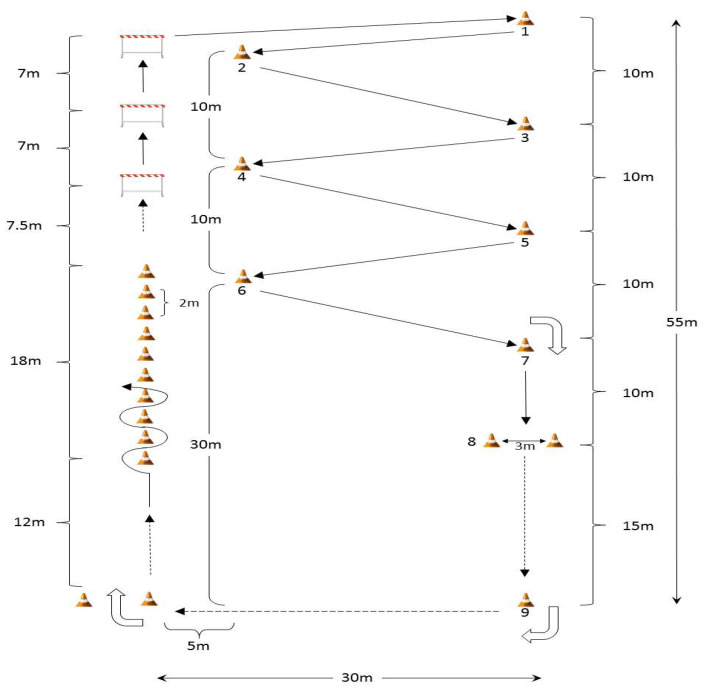
Illustration of the Hoff circuit exercise. The participants dribble the ball through the circuit. The circuit length is set of 51.5 m on left-hand side and 55 m on the right-hand side and the width is set of 35 m. The participants were required to perform backward dribbling between the cone 7 to the gate 8. Three hurdles (30–35 cm height) and 22 cones (two for the starting line and two cones for the backward run gate) were set on the field. Total distance of per lap is 290 m. The distance between hurdle 3 and cone 1 is 30.5 m. The distance between cones (1–2, 2–3, 3–4, 4–5, 5–6 and 6–7) is 25.5 m each [27].

**Figure 3 children-09-00907-f003:**
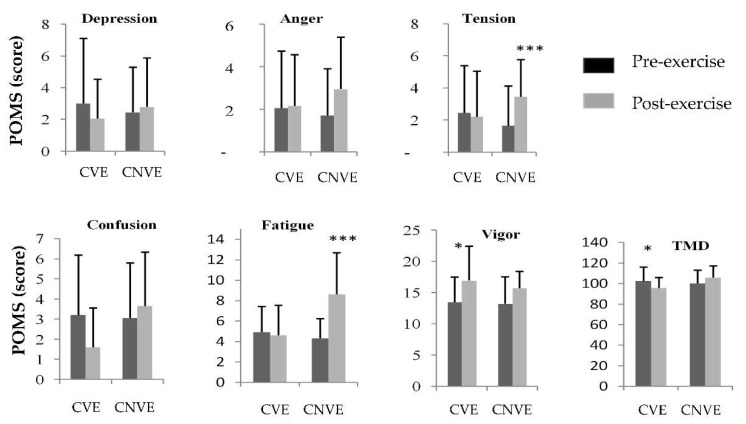
The profile of mood states (POMS) scores during the Hoff circuit exercise with and without verbal encouragement during pre-exercise and post-exercise measures. CVE, circuit exercise with verbal encouragement; CNVE, circuit exercise without verbal encouragement; TMD, total mood disturbance. Error bars indicate within-subject standard deviation. * a significant difference between pre-exercise and post-exercise values. * *p* < 0.05, *** *p* < 0.001.

**Table 1 children-09-00907-t001:** Comparison of the distance traveled, Hrmax, and RPE variables between circuit exercise with and without verbal encouragement.

Variables	CVE	CNVE	*d*	Rating
Distance traveled (m)	1688.1 ± 206.7	1483.7 ± 180.7 ***	1.08	Large
Hrmax (%)	88.31 ± 2.45	84.06 ± 2.27 ***	1.86	Large
RPE (AU)	7.5 ± 0.82	6.3 ± 0.97 ***	1.37	Large

Hrmax: maximal heart rate; RPE: rating of perceived exertion; CI 95%, d: Cohen’s coefficient, CVE: circuit exercise with verbal encouragement, CNVE: circuit exercise without verbal encouragement, *** *p* < 0.001.

**Table 2 children-09-00907-t002:** Analysis of Variance results with 2 × 2 repeated measures (exercise method: circuit with verbal encouragement and circuit without verbal encouragement) × effort: pre- and post-exercise).

Variables	Effort (Main Effect)		Exercise Method (Main Effect)		Interaction	
	F (1,19)	η^2^	F (1,19)	η^2^	F (1,19)	η^2^
**Depression**	0.012	0.001	0.40	0.02	1.22	0.06
**Anger**	0.18	0.009	2.20	0.11	1.23	0.06
**Anxiety**	0.08	0.005	2.66	0.12	8.19 **	0.30
**Confusion**	2.48	0.11	0.99	0.05	3.11	0.14
**Fatigue**	9.87 ***	0.34	8.94 ***	0.32	9.94 ***	0.34
**Vigor**	0.66	0.03	8.55 ***	0.31	0.21	0.01
**TMD**	1.57	0.07	6.02 *	0.24	4.27 *	0.18

TMD: total disturbance of mood. * *p* < 0.05, ** *p* < 0.01, *** *p* < 0.001.

## Data Availability

The data presented in this study are available on request from the corresponding author.

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
