# Peer review of "The Effects of Verbal Encouragement during a Soccer Dribbling Circuit on Physical and Psychophysiological Responses: An Exploratory Study in a Physical Education Setting"

_children, 2022, doi:10.3390/children9060907_

Round 1

Reviewer 1 Report

-          Thank you for the opportunity to review this interesting study. For the most part, it is generally well written, however there are specific points of clarification required to allow replication and transparency.

-          Line 24 is missing the word contributed

-          Line 35 – I don’t think the additional (d) should be in brackets

-          I think a line of rationale in the abstract would be useful to provide some context/motive for the study.

-          The introduction is clear and concise and sets out the main issues to be addressed. I am not completely convinced by the rationale – perhaps a stronger justification for the study and the implications would be helpful.

-          Line 84 – formatting error with a full stop before Participants.

-          Can you include a power analysis? How representation of the population were your sample? What were habitual levels of MVPA/sport participation?

-          Typo on line 97 – should this read “Together”?

-          I suggest you revise the order of the information in the participants section to discuss ethics, then recruitment procedures and then the sample. You could then move the information which discusses more of the experimental procedure (line 89 to 93) to the procedure section.

-          Please have materials/data collection before procedure to improve readability.

-          Line 112 – sorry to be picky but did you assess individual circadian rhythm? Without assessing it you cannot say if having the session avoided variations as some participants may be morning types and some evening. If no assessment was done then please remove this section and simply state that the sessions took place at the same time (delete “to avoid circadian rhythm variations”).

-          Please include more specific detail in the procedure about who collected the data and how it was completed – did students complete the POMS online or on paper, for example? Did they complete in a classroom, or outside where they completed the circuit training? Was the person responsible for data collection (e.g. skin folds) suitably qualified?

-          Figure 1 is challenging to navigate. This may be due to alignment issues but it is hard to work out what the two representations for week 1 show and how they map onto sessions. Please revise it for clarity.

-          Line 155-158 would be better located in the rationale for the study.

-          Please reference the Vameval test procedure. This same task features in a number of test batteries so please be specific so readers are aware of variations.

-          Please justify the time of administration of the POMS –why 5 mins after and not 10 mins, for example?

-          please reference the reliability information for all measures.

-          Please be more precise in reporting of results. For example, all means and SD should be reported as they are not easily interpretable from the figures.

-          I am not convinced figure 3 is necessary if the data is reported accurately.

-          Line 279 – please spell out SSG at first mention.

-          Line 291 – please add name of author for reference 15 in text – same comment on line 298 and 320.

-          Line 310 – I think it would be more appropriate to say results show participants run a further distance, rather than the longest distance possible as this is subjective unless they ran absolute max for the task.

-          It would be useful to map your findings on to relevant theory in the discussion.

-          I think the conclusion as it stands is quite weak. Please broaden this out to discuss the implications and future directions of the work. The “so what” question needs to be answered here as this is currently lacking in the manuscript.

Author Response

Dear Reviewer:
Thank you very much for your suggestions and the valuable comments in the second round of review. We have revised/improved the manuscript according to your comments. We provide point-by-point responses (in RED color) to the reviewers’ comments below. Revision in responses to each question/concern raised by the reviewers are noted with tracked changes in the manuscript.

REVIEWER 1  

Thank you for the opportunity to review this interesting study. For the most part, it is generally well written, however there are specific points of clarification required to allow replication and transparency.
Responses: Thanks for your time and efforts. Your comments strengthen the quality of the manuscript. We fully addressed the point you highlighted throughout the manuscript.

Line 24 is missing the word contributed
Responses: Done.

Line 35 – I don’t think the additional (d) should be in brackets.
Responses: Done.

I think a line of rationale in the abstract would be useful to provide some context/motive for the study.
Responses: Thanks for your suggestion. Please see the changes at the beginning of the abstract “Verbal encouragement (VE) can be used by physical education practitioners for boosting motivation during exercise engagement. For this reason, this study aimed to examine the effect of verbal encouragement on physical and psychophysiological responses in a physical education (PE) context.”

The introduction is clear and concise and sets out the main issues to be addressed. I am not completely convinced by the rationale – perhaps a stronger justification for the study and the implications would be helpful.
Responses: We have strengthened the rationale of study and justification to conduct this study has been added. Please refer to Line 76-87

Line 84 – formatting error with a full stop before Participants.
Responses: Done.

Can you include a power analysis? How representation of the population were your sample? What were habitual levels of MVPA/sport participation?
Responses: We added a sample size estimation between Line 105-109 to state the power analysis. “A priori power analysis was calculated by using the G*Power software (Version 3.1.9.4., Düsseldorf, Germany), based on the t test family (Means: difference between two dependent means). The analysis output showed that a sample size of 19 subjects would be sufficient to find significant differences (effect size = 0.887, power (1-ß) = 0.95 with an actual power of 95.46 in this study.”

Regarding to the representation of the studying population, we indeed recruited school male students with healthy conditions. However, we failed to collect the data by using proper questionnaire (such as IPAQ) to examine their daily physical activities during the participations. Therefore, we are unable to state their moderate to vigor physical activity during study period. We will take this point as a methodological concern in our future study. Thanks you very much.

Typo on line 97 – should this read “Together”?
Responses: Done.

I suggest you revise the order of the information in the participants section to discuss ethics, then recruitment procedures and then the sample. You could then move the information which discusses more of the experimental procedure (line 89 to 93) to the procedure section.
Responses: Revised accordingly. 

Please have materials/data collection before procedure to improve readability.

Responses: Revised accordingly.

Line 112 – sorry to be picky but did you assess individual circadian rhythm? Without assessing it you cannot say if having the session avoided variations as some participants may be morning types and some evening. If no assessment was done then please remove this section and simply state that the sessions took place at the same time (delete “to avoid circadian rhythm variations”)
Responses: Done.

Please include more specific detail in the procedure about who collected the data and how it was completed - did students complete the POMS online or on paper, for example? Did they complete in a classroom, or outside where they completed the circuit training? Was the person responsible for data collection (e.g. skin folds) suitably qualified?
Responses: We added a sentence to state the person conducted the skinfold assessment was experienced. “A well-trained sports scientist conducted the anthropometric measurements in this study.”

We also added a sentence to state how the participants completed the POMS assessment. “The participants completed the POMS on papers outside where they completed the circuit exercise.”

Figure 1 is challenging to navigate. This may be due to alignment issues but it is hard to work out what the two representations for week 1 show and how they map onto sessions. Please revise it for clarity.
Responses: Revised accordingly.

Line 155-158 would be better located in the rationale for the study.

Responses: Done.

Please reference the Vameval test procedure. This same task features in a number of test batteries so please be specific so readers are aware of variations.
Responses: The reference of [12] for the Vameval test in our study was added.

Selmi, O., Khalifa, W. B., Ouerghi, N., Amara, F., & Zouaoui, M. Effect of verbal coach encouragement on small sided games intensity and perceived enjoyment in youth soccer players. J. Athl. Enhanc.2017, 3, 16-7.

Please justify the time of administration of the POMS –why 5 mins after and not 10 mins, for example?
Responses: We chose 15 min before and 5 min after according to Selmi et al., 2018 (Selmi, O., Haddad, M., Majed, L., Khalifa, B., Hamza, M., Chamari, K. Soccer training: high-intensity interval training is mood disturbing while small, sided games ensure mood balance. J. Sports. Med. Phys. Fit. 2018, 58 (7-8), 1163-1170).

please reference the reliability information for all measures.
Responses: Added reliability information for Vameval test, POMS, and PACES.

Please be more precise in reporting of results. For example, all means and SD should be reported as they are not easily interpretable from the figures.

Responses: Revised accordingly.

I am not convinced figure 3 is necessary if the data is reported accurately.

Responses: We remove the figure 3 to avoid the confusion.

Line 279 – please spell out SSG at first mention

Responses: Revised as a soccer dribbling circuit exercise.

Line 291 – please add name of author for reference 15 in text – same comment on line 298 and 320.

Responses: Done.

Line 310 – I think it would be more appropriate to say results show participants run a further distance, rather than the longest distance possible as this is subjective unless they ran absolute max for the task.

Responses: Revised accordingly.

It would be useful to map your findings on to relevant theory in the discussion.

Responses: Done. We have checked the discussion and revised the readiness in the revision.

I think the conclusion as it stands is quite weak. Please broaden this out to discuss the implications and future directions of the work. The “so what” question needs to be answered here as this is currently lacking in the manuscript.

Responses: Thank you the suggestion. The conclusion was revised as your concerns.

Reviewer 2 Report

This manuscript presents a relevant topic to publish in Children, which could be accepted with some minor revisions. 

In my opinion, the introduction provides adequate information and structure to set up the research questions raised in the manuscript; the methodology provides sufficient detail; the results section is sufficiently clear and precise; the discussion of results is based on previous literature;

  • Some aspects of formatting should be corrected (spelling). Please, correct what is pointed out in the body of the manuscript; 

Author Response

Dear Reviewer:
Thank you very much for your suggestions and the valuable comments in the second round of review. We have revised/improved the manuscript according to your comments. We provide point-by-point responses (in RED color) to the reviewers’ comments below. Revision in responses to each question/concern raised by the reviewers are noted with tracked changes in the manuscript.

This manuscript presents a relevant topic to publish in Children, which could be accepted with some minor revisions. 
Responses: Thanks for your time and efforts. Your comments strengthen the quality of the manuscript. We fully addressed the point you highlighted throughout the manuscript.

In my opinion, the introduction provides adequate information and structure to set up the research questions raised in the manuscript; the methodology provides sufficient detail; the results section is sufficiently clear and precise; the discussion of results is based on previous literature.
Responses: Thank you for the positive feedback.

Some aspects of formatting should be corrected (spelling). Please, correct what is pointed out in the body of the manuscript; 
Responses: Thank you very much, the manuscript has been revised according to your remark. Please see the changes in the revision.
